# Predicting innovative firms using web mining and deep learning

**Jan Kinne**[1,2,3]☯*, **David Lenz**[ID][3,4]☯

**1** Department of Economics of Innovation and Industrial Dynamics, ZEW Centre for European Economic Research, Mannheim, Germany, **2** Department of Geoinformatics Z_GIS, University of Salzburg, Salzburg, Austria, **3** istari.ai, Mannheim, Germany, **4** Department of Econometrics and Statistics, Justus-Liebig-University, Gießen, Germany

☯ These authors contributed equally to this work.
* jan.kinne@zew.de

**Data Availability Statement:** The data underlying this study comes from the following two main datasets: The Mannheim Enterprise Panel (MUP), a firm database including all economically active firms in Germany, and the Mannheim Innovation Panel (MIP), a database with firms sampled from

## Abstract

Evidence-based STI (science, technology, and innovation) policy making requires accurate indicators of innovation in order to promote economic growth. However, traditional indicators from patents and questionnaire-based surveys often lack coverage, granularity as well as timeliness and may involve high data collection costs, especially when conducted at a large scale. Consequently, they struggle to provide policy makers and scientists with the full picture of the current state of the innovation system. In this paper, we propose a first approach on generating web-based innovation indicators which may have the potential to overcome some of the shortcomings of traditional indicators. Specifically, we develop a method to identify product innovator firms at a large scale and very low costs. We use traditional firm-level indicators from a questionnaire-based innovation survey (German Community Innovation Survey) to train an artificial neural network classification model on labelled (product innovator/no product innovator) web texts of surveyed firms. Subsequently, we apply this classification model to the web texts of hundreds of thousands of firms in Germany to predict whether they are product innovators or not. We then compare these predictions to firm-level patent statistics, survey extrapolation benchmark data, and regional innovation indicators. The results show that our approach produces reliable predictions and has the potential to be a valuable and highly cost-efficient addition to the existing set of innovation indicators, especially due to its coverage and regional granularity.

## Introduction

Innovations can disrupt individual industries with game-changing technology and the most radical innovations can even reshape whole economies. Despite having a destructive element, innovation is widely considered to be a main driver of long-term economic growth. Such growth may be kick-started by radical innovations or driven forward by a constant stream of so called incremental innovations which cause continuous change. Measuring and promoting innovation is the main objective of STI (science, technology and innovation) policy, which

the MUP and surveyed in the context of the German Community Innovation Survey (https://www.zew.de/PJ345-1). The MUP is based on data gathered by the private company Creditreform. Given this is proprietary data owned by these privately held companies, we are legally not allowed to distribute any datapoints from the dataset. The MIP on the other hand contains highly confidential information on company turnover, product margins and so on, that has been provided by the companies in the context of the innovation survey. The data is available to the public in the context of a GDPR compliant setting. Hence, it can be accessed by researchers via the Research Data Center of the ZEW Centre for European Economic Research: https://kooperationen.zew.de/en/zew-fdz/home.html We can confirm that we did not receive any special privileges in accessing the data, except for the omission of having to conduct our calculations in a specifically protected IT environment within the ZEW Research Data Center. This is due to the substantial legal enclosure of author JK as an employee of ZEW. As such, JK is able to access the data directly, without having to register access for the ZEW Research Data Center. Hence, other scientists are eligible for accessing exactly the same Mannheim Innovation Panel data. Concerning access to the Mannheim Enterprise Panel (MUP) dataset, the proprietary and commercial basedata of Creditreform/Bureau van Dijk can be accessed by all researchers as well. However, access costs money: https://www.bvdinfo.com/en-us/our-products/data/international/orbis.

**Funding:** The German Federal Ministry of Education and Research provided funding for the research project (TOBI - Text Data BasedOutput Indicators as Base of a New Innovation Metric; funding ID: 16IFI001, Prof. Dr. Peter Winker) (https://www.bmbf.de/en/index.html). Additionally, DL and JK founded and are employed by istari.ai. These funders had no role in study design, data collection and analysis, decision to publish, or preparation of the manuscript. The specific roles of these authors are articulated in the 'author contributions' section.

**Competing interests:** The authors DL and JK founded and are employed by istari.ai. Istari.ai develops a "InnoProb" innovation prediction service based on the methodology presented in this article. This does not alter our adherence to PLOS ONE policies on sharing data and materials.

requires an accurate and timely picture of the current state of the STI system to implement policy measures in an evidence-based manner. However, traditional innovation indicators from questionnaire-based surveys or patent data struggle to provide the full picture [1–3]. [4] identified shortcomings of traditional innovation indicators concerning their coverage, granularity, timeliness, and cost. The authors proposed to use firm websites as a source of firm-level innovation indicators, leveraging the fact that almost all relevant firms have websites nowadays. Websites are used as platforms to provide information on a firm's products and services, achievements, strategies, and relationships. All these aspects may be related to innovations developed by the firm. Innovation, in this context, is defined as the introduction of a new or significantly improved product or process [5]. Most of the information on websites is codified as text and extracting innovation-related information from these web texts and transferring this information into a reliable firm-level innovation indicator is the aim of this study.

Text mining algorithms can be used to extract knowledge from large document collections and turn them into valuable economic information [6–10]. As a result, text mining became one of the most promising approaches in economic analysis to provide novel tools and insights to economists. At the methodological level, great progress has been made in natural language processing (NLP), driven by the rapid increase in computational power and the availability of large text corpora [11]. Especially artificial neural networks have shown very promising results when used for the classification of text documents into certain categories [12, 13].

In this study, we use information from the Mannheim Innovation Panel (MIP), a questionnaire-based innovation survey of firms, to label the websites of surveyed firms as associated to either a product innovator firm or a non-innovator. This labelled data set is then used to train a deep neural network to predict the probability of firms to be product innovators based solely on their website text. Fig 1 outlines our proposed approach. The predicted product innovator probabilities can be interpreted as a continuous firm-level indicator of innovation.

We assess our proposed approach using the following two research questions:

- **Research Question 1**: Can deep neural networks be used to reliably identify product innovator firms solely based on their website texts?

- **Research Question 2**: Are the resulting firm-level, regional, and sectoral patterns from such a prediction model similar to the patterns observed from established innovation indicators when the model is applied to a large out-of-sample dataset of firm website texts?

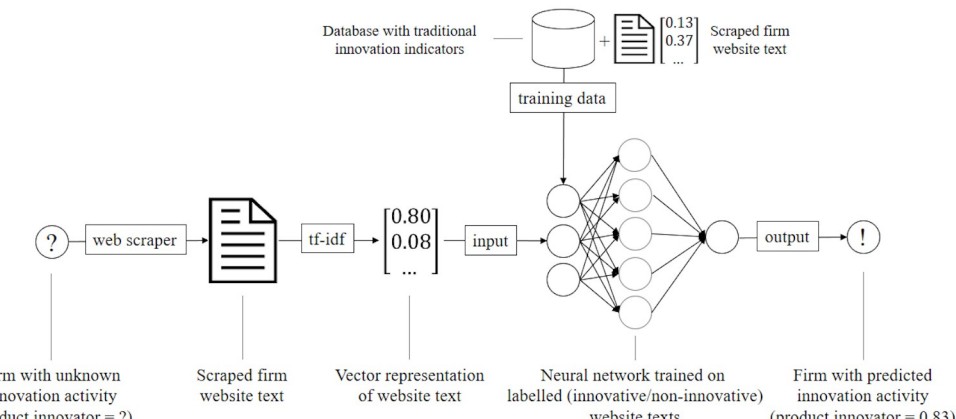

**Fig 1. Proposed product innovator prediction framework.** Web scraped texts from firms with a known innovation status are used to train a artificial neural network. The trained model can then be used to predict the innovation status of out-of-sample (unlabelled) firms based on their website texts.

The remainder of this paper is structured as follows. First, we present our data, followed by our methods. Our results section is twofold. In the first part, we present the results of our artificial neural network classification model. In the second part, we apply our classification model to a large dataset of German firm websites and assess the resulting firm-level, sectoral, and regional patterns. The results are discussed and summarized in the last two sections.

## Data

In the following section we present our base datasets, the Mannheim Enterprise Panel (MUP), a firm database containing all economically active firms in Germany, and its derivative, the Mannheim Innovation Panel (MIP), which is an annual innovation survey and the German contribution to the European-wide Community Innovation Survey (CIS). We also present how we obtained texts from the websites of firms in both the MUP and MIP using ARGUS web scraper [14].

### MUP firm dataset

The Mannheim Enterprise Panel (MUP) is a panel database which covers the total population of firms located in Germany. It contains more than three million firms which are updated on a semi-annual basis. The data also includes firm characteristics such as the industry (NACE codes; a classification of economic activities in the European Union), postal addresses, number of employees, as well as the website address (URL) of the firms (see [15]).

For our analysis, we use MUP data restricted to firms that were definitely economically active in early 2018. The resulting dataset contains 2.52 million firms and 1.15 million URLs (*URL coverage* of 46%). A prior analysis of this dataset by [4] showed that URL coverage differs systematically with firm characteristics. Only a fraction of very young (younger than two years) and very small firms (fewer than five employees) are covered by a URL after controlling for the search quality of the data provider. Sectoral and regional differences can be expected as well. Some regions (especially those with low broadband Internet availability and a low population density) and some sectors, like agriculture, exhibit lower URL coverage. However, given that most of the innovation activity is conducted by middle-sized and larger firms [16], which are well covered in the dataset, the dataset can be assumed to be suitable to analyze the German innovation system.

### MIP innovation survey data

The Mannheim Innovation Panel (MIP) is an annual questionnaire-based innovation survey of firms sampled from the MUP database. The survey is designed as a panel survey, such that firms in the sample are surveyed every year. Firm closures and mergers are substituted by randomly sampled additional firms every two years. The MIP is the German contribution to the Community Innovation Survey (CIS), which is conducted every two years in the European Union. CIS data has been used as base data in an array of studies [17]. The survey methodology and definition of innovation follows the "Oslo Manual" [5] and covers firms with five or more employees in manufacturing and business-oriented services. Each year, firms are asked whether they introduced new or significantly improved products ("product innovations") during the three years prior to the survey. The firms receive detailed and sector-specific descriptions with examples of product innovations to help them filling the survey. In our study, we use the firms' status as product innovators (yes/no) as the binary target variable.

Each MIP survey, following the CIS guidelines, covers a three-year reference period (the three years preceding the year the survey is conducted). The most recent survey available to us is 2017 which covers the years 2014, 2015, and 2016. Given that we use web texts scraped from

the firms' websites in 2018 matched to the 2017 MIP survey, some firms may have changed their innovation status during this time lag. A relevant fraction of firms actually change their status as product innovators between years, such that they may be an innovator in 2015 but not in 2016 [18]. We decided to cope with this issue by restricting our analysis to firms that had a "stable" innovation status in the surveys of 2015, 2016, and 2017 (covering the years 2012 to 2016). This means they replied to be product innovators in either all of these years or none of these years. It can be assumed that such firms are less likely to have changed their innovation status between the survey of 2017 and our test day in 2018. This approach reduces our sample of MIP 2017 firms from 18,062 to 4,481.

The sampling procedure of the survey (oversampling of industries and size classes where innovation in more prevalent) results in an over-representation of innovative firms in the MIP. As a result 32% of the firms in our database are product innovators. Projected to the overall firm population in Germany, the share of product innovators can be expected to be in the range of 27% for the target population in the MIP [19]). This oversampling of innovative firms and a restriction to firms with at least five employees results in a dataset, in which firms are larger, in terms of their number of employees, than firms in the overall firm population in Germany. The mean number of employees in our final MIP sample is 277.5 and the median is 23. Very young firms cannot be included in our sample, as firms have to have taken part in the survey at least three times. Nevertheless, we are able to show that the restriction to firms with a "stable" innovation status results in a higher quality of our training data (see Appendix).

## ARGUS website texts

We used ARGUS [4], a free web scraping tool based on the Scrapy Python framework, to scrape texts from the websites of all firms in both our MUP and MIP datasets. We used ARGUS simple language selection heuristic (set to German), which was shown to help limiting the downloaded texts to a certain language [4]. According to [4], about 90% of the webpages downloaded this way can be expected to be in German, with some sectors, like the pharmaceuticals and mechanical engineering sector, exhibiting higher shares of non-German webpages (most of them in English).

Webpages are not downloaded randomly from the firms' websites. Instead, ARGUS starts at a firm's main page ("homepage") and then continues downloading those sub-webpages with the shortest URL. The rationale is that more general information on the firm is available at the top level of its website (e.g. "firm-name.com/products", "firm-name.com/team"), which should be given priority over more specific information (e.g. "firm-name.com/news/2017/august/most_read"). This top-level approach is intended to capture texts that represent firm-level business activity profiles instead of specific product-level descriptions which are found on low-level webpages. Even though the latter may inform about individual product innovations, we assume that a top-level business profile description may allow our prediction model to learn combinations of more general signal words that reliably predict product innovator firms, regardless of what exact product innovations they implemented. Such a generalization would be desirable, especially as our training data consists of firms from both ends (consistently innovative or consistently non-innovative) of the overall firm distribution.

The number of downloaded webpages per firm website is defined by a limit parameter in ARGUS, which was set to 25 for this analysis. Hereby, we follow the recommendations of [4] who found that 50% of all firm websites can be scraped completely when this limit parameter is set to 15. Reaching 90% would require raising this threshold to 250, highlighting that web-based studies have to deal with outlier websites, as some firms (especially large ones) have massive websites with ten-thousands of webpages. Following the suggested practice by [4], we

**Table 1. Firms in datasets after filtering steps.**

| dataset | base data | unstable innovation | no URL | errors/redirects | final data |
|---------|-----------|---------------------|--------|------------------|------------|
| MUP | 2,523,231 | N/A | -1,374,383 | -463,791 | 685,057 |
| MIP | 18,062 | -13,587 | -456 | -893 | 3,126 |

excluded websites which redirect to a different domain when requesting their first webpage (i.e. homepage). A practice which should ensure that crawled websites belong to the corresponding firms. This was also shown to not result in any sectoral or firm age selection bias. Table 1 presents our data after excluding such initial redirects and download errors caused by non-existing websites.

During web scraping, all texts found on a firm's website are downloaded, regardless of their content and relevance to the study. To filter out unwanted (*bloat*) sub-webpages, we applied an intermediate filtering step which is described in the Appendix.

## Methodology

In this section, we present how the website texts were preprocessed and transferred to term frequency–inverse document frequency (tf-idf) vectors. Tf-idf represents documents as a fixed size vector by counting words in each document (term frequency) and weighting each frequency by the inverse of the term's overall document frequency. We then describe the architecture of our deep neural network model for binary text classification, and how we evaluate the model's classification performance.

### Web text preprocessing

We reduced the preprocessing of our texts to a minimum. The scraped web texts were standardized to lowercase and all characters not in the German alphabet were removed (keeping *Umlaut* special characters, whitespaces, and ampersands). Tests with word stemming procedures, which reduce words to their stem (e.g. "innovation" and "innovator" to "innovat"), did not increase our classification performance and we refrained from using it.

### Web texts as numerical tf-idf vectors

We used the term frequency–inverse document frequency (tf-idf) scheme to represent the website texts as sparse vectors (see e.g. [20]). The tf-idf algorithm transfers each document to a fixed size sparse vector of size $V$, where $V$ is the size of a dictionary composed of all words found in the overall text corpus. We restricted our dictionary to words with a minimum document frequency of 1.5% and a maximum document frequency of 65% (*popularity based filtering*), resulting in a dictionary size $V$ of 6, 144 words. Each entry in the tf-idf vector of a document corresponds to one word in the dictionary, representing the relative importance of this word in the document (i.e. the website). Words that do not appear in a given document are represented by a 0 value. Specifically, in a first step (the tf step) the number of appearances per word in a single document are counted. In a second step, the inverse document frequency (idf) is used as a weighting scheme to adjust the tf counts. Conceptually, the idf weights determine how much information is provided by a specific word by means of how frequently a word appears in the overall document collection. The intuition is that very frequent words that appear in a lot of documents, should be given less weight compared to less frequent words, as infrequent words are more useful as a distinguishing feature.

## Web text classification with a deep neural network

Deep neural networks showed remarkable success when applied as text classification models [12, 13]. Different deep neural network architectures were proposed and showed varying performance in different NLP tasks. We tried several neural network architectures (convolutional neural networks, recurrent neural networks, both with long short-term memory and gated recurrent units) and also compared their performance in our specific classification task with more traditional models (naive Bayes classifier, logistic regression, decision trees). In this iterative process, an architecture that could be described as a *undercomplete autoencoder-like neural network* turned out to be the model with the best classification performance. Autoencoder-style neural networks (see e.g. [21]) impose a "bottleneck" (hidden layers with very few neurons) in the network architecture which are intended to force the learning of a highly compressed representation of the network's input. While the output of a standard autoencoder network has the same dimensionality as the network's input, the output of an undercomplete autoencoder network has a smaller dimension than the network's input.

Our final network consists of four hidden layers with intermediary dropout layers, which are intended to improve the network's generalization by ignoring (*dropping*) neurons during the training phase [22]. The network's first hidden layer consists of 250 neurons, the following two hidden layers consist of only five neurons each (the "bottleneck"), while the forth and last hidden layer contains 125 neurons. We used *scaled exponential linear units* (SELU, [23]) as activation functions in the hidden layers. The network's output layer consists of a single neuron with a *sigmoid* activation function, a common approach to obtain an output between 0 and 1 from a neural network in binary classification tasks (see e.g. [20]). We used the common Adam optimizing algorithm [24] for the stochastic optimization of the network weights.

## Results

In this section, we use the dataset of MIP firms with a surveyed innovation status to train our product innovator classification model and test the model's performance using a retained part of the training data (the test set). In the second part of this section, we apply the innovation prediction model to about 700,000 firms from the MUP to predict whether they are product innovators and then examine the resulting firm-level, sectoral, and regional patterns.

### Product innovator prediction model performance

After filtering bloat webpages from our MIP dataset (see Appendix), we aggregated all remaining webpages to the firm level, keeping only the first 5,000 words per firm. As a result, each firm is represented by a single document with a maximum length of 5,000 words from non-bloat webpages only. We randomly selected 75% of the data as training set and retained 25% as test set. Table 2 details precision, recall, f1-score, and support for the resulting classifier applied to the test set (classification threshold for the probabilities of 0.5). If the model classifies a firm as a product innovator, it is correct in roughly 4 out of 5 cases, as it can be seen by the 81% precision for the product innovator class. The model retrieves 64% of all product innovator firms

**Table 2. Product innovator classification report for test set.**

| label | precision | recall | f1-score | support |
|---|---|---|---|---|
| non-innovator | 0.81 | 0.91 | 0.86 | 429 |
| product innovator | 0.81 | 0.64 | 0.71 | 255 |
| avg / total | 0.81 | 0.81 | 0.80 | 684 |

and 91% of all non-innovator firms in the test dataset (recall). The overall f1-score of the model is 80%.

We also tested the classification model using two alternative configurations: a first model that takes a vector with only the firms' (normalized) age, number of employees, and (*one-hot* encoded) sectors as input and a second model for which we concatenated these firm characteristics vectors and the corresponding tf-idf text vectors. For both alternative models we did not alter the original model's architecture, except for changing the number of neurons in the input layer in order to allow for the input of the new vectors. It turned out that the original text-only model outperforms the firm characteristics-only model by about ten percentage points in overall precision, recall, and f1-score. Most notably, the text-only model showed a twenty percentage points higher recall in the already difficult to detect product innovator class. Furthermore, we found that the alternative text plus firm characteristics model does not show an improved performance over the text-only model.

## Out-of-sample product innovator predictions

We used the trained model to predict product innovator probabilities for 685,057 MUP firms. The resulting distribution of product innovator probabilities is shown in Fig 2. The mean probability is 0.253 and the median is 0.203. The lowest predicted probability is 0.029 and the highest is 0.944.

## Comparison to firm-level patent statistics

We obtained firm-level patent statistics (a long-time established indicator for innovation in firms, see e.g. [25, 26]) for 2017 from the European Patent Office. Because patents vary greatly in their importance from sector to sector, we limited our comparison to sectors where at least five percent of the companies are patent holders (mechanical engineering, electronic products, petrochemistry, pharmaceuticals, metal product, other products, and materials). We also excluded patents that were filed prior to 2006 (ten years is the average lifetime of a patent in our database) to account for the decreasing economic and technological value of aging patents.

The significant Spearman's correlation between our predicted product innovator probabilities and the firms' statuses as patent holders is 0.29. The correlation with the firms' number of patents is 0.25. We also ran two regression analyses to control for sector and size effects in

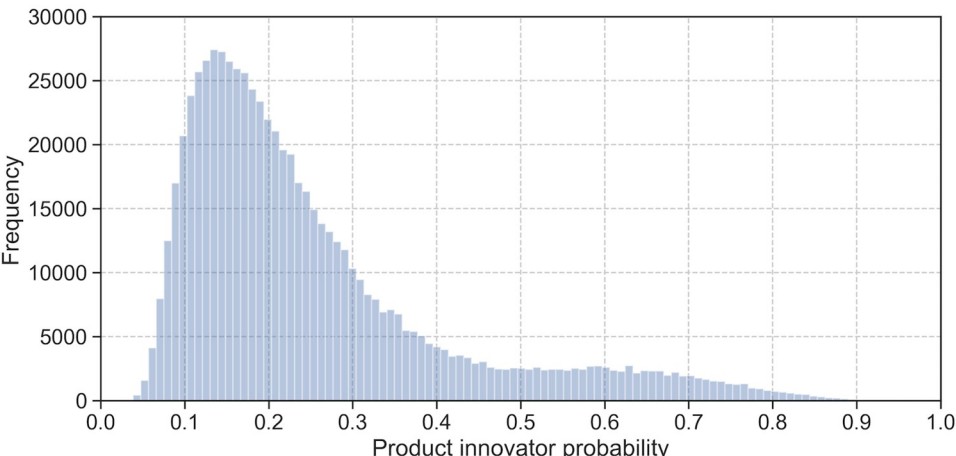

**Fig 2. Product innovator probabilities distribution.** Histogram of predicted product innovator probabilities for 685,057 firms.

**Table 3. Firm-level patent statistics regression results.**

| Dependent variable | Patent holder | Patent count |
|---|---|---|
| | *Variable of interest* | |
| **Product innovator probability** | 0.265*** | 35.441*** |
| | *Controls and constant* | |
| **Sector** | Yes | Yes |
| **Employees (log)** | 0.054*** | 2.274*** |
| **Constant** | -5.221*** | 0.003*** |
| | *Model statistics* | |
| **Regression model type** | Robust logit (average marginal effects) | Robust Poisson (incident rate ratios) |
| **Observations** | 35,291 | 35,291 |
| **Pseudo R-squared** | 0.24 | 0.52 |
| **Wald chi2/F-test** | 4,653.30*** | 911.10*** |

these correlations. The results presented in Table 3 indicate that our predicted product innovator probabilities are strongly related to the probability of a company to hold a patent and having a high patent count. We need to emphasize that this regression analysis is meant to provide a framework for a controlled correlation analysis and the results should not be interpreted as indicating causality.

## Comparison to survey extrapolation statistics

To sort the firms into one of the two classes of product innovators and non-innovators (to make them comparable to existing sector and size level survey benchmark data), we are required to set a rather arbitrary product innovator probability classification threshold. Setting this threshold to 0.5 (the same that was used during the model's training) results in 10.31% of the MUP firms being classified as product innovators. Validating this number is difficult as there is no available reference value for the share of product innovator firms in the overall firm population of Germany because MIP survey data can only be used to extrapolate representative shares for that part of the firm population which is covered by the survey (firms with five or more employees in manufacturing and business-oriented sectors) [19]. As a result, we can benchmark our prediction results against extrapolated MIP data only in these sectors and size groups, which corresponds to 89,372 of the MUP firms.

For this subgroup, the predicted share of product innovators is 21% (classification threshold of 0.5) while the MIP survey extrapolations indicate a higher share of 27%. This underprediction of product innovators can be related to our model's rather low recall of the product innovator class (see previous section). To adjust for this discrepancy, we decided to calibrated our classification threshold to a value that produces the same number of innovative firms anticipated by the survey extrapolation benchmark of 27%. This calibrated classification threshold of 0.401 was subsequently used to label all 685,057 MUP firms as either product innovators or non-innovators. Naturally, this resulted in an increased share of product innovators in the overall firm population from 10.31% to 15.12%.

In the Appendix, breakdowns by sectors and size classes can be found. Concerning sectors, they show that even though the overall trend and the proportions between sectors are similar to the survey benchmark, underprediction can be seen in all sectors except for wholesale, consulting, and especially ICT firms. Concerning size classes, it can be seen that our predictions match the survey benchmark very well, except for very large firms with more than 1,000 employees where we underestimate the share of product innovators.

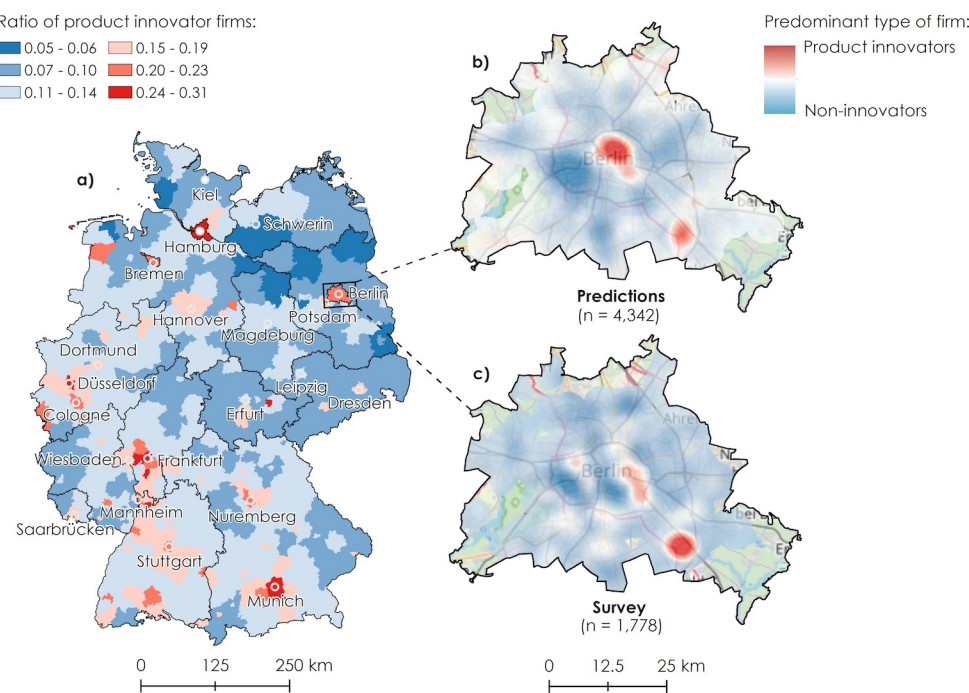

**Fig 3. Geographic pattern of product innovator firm predictions.** Predicted share of product innovator firms in local firm population by German districts (a). Microgeographic pattern of product innovator firms in Berlin, predicted (b) and surveyed (c). Basemap: OpenStreetMap.

## Geographic patterns

Map *a* in Fig 3 shows the predicted ratio of product innovator firms to all firms for 432 German districts. It can be seen that city districts exhibit higher shares of product innovator firms. This fact is confirmed by a high and significant Spearman's correlation coefficient of 0.61 between district population density (a proxy for urbanity) and the ratio of product innovator firms. It can also be seen that the vicinities of some major agglomeration areas in the South-West of Germany (Munich, Nuremberg, Stuttgart, Mannheim, and Frankfurt) exhibit high shares of product innovator firms as well. We also find positive and significant correlations between the local share of predicted product innovator firms and the share of the local population working as research and development staff (0.72) as well as with the number of local high-tech patent application per one million inhabitants (0.67). Both these statistics were obtained from the European Statistical Office at the level of NUTS-2 regions for the most recent years available.

The detailed address information in our MUP firm database allows us to disaggregate the geographic pattern as it is shown in map *b* of Fig 3 and to analyze individual regions at a microgeographic level. For the German capital of Berlin, a special survey of the MIP is conducted every year [27], covering a high proportion of firms from manufacturing and business-oriented services in Berlin (the data from this survey was not used for the training of our prediction model). This comprehensive dataset allows us to map the density of our predicted product innovators in Berlin against the observed density from the MIP special survey (map *c* in Fig 3). For map *b* which shows our prediction results, we selected firms that are from sectors and size classes covered in the survey (4,342 of 35,998 firms in Berlin). Map *c* shows the same pattern for 1,778 firms that answered the innovation survey questionnaire. Both firm location

patterns were used to calculate kernel density maps using the same set of parameters. It can be seen that the two densities resemble each other in their overall appearance, with major hot-spots in the eastern city center of Berlin (city districts of Mitte, Prenzlauer Berg, and Friedrich-shain-Kreuzberg) as well as in the area around Adlershof (a major science and technology park) in the South-East.

## Discussion

In this paper, we introduced a deep learning approach to predict product innovator firms based on their website texts. With our deep learning classification model we achieved an overall f1-score of 0.80 in the test dataset, which indicates a very good but by no means perfect prediction performance. While we found that precision is balanced for both classes, the model performed less good concerning the recall of the "product innovator" class. Further performance improvements regarding the latter are therefore desirable and could be achieved with larger training data or an improved model architecture, for example by relying on recent developments concerning pre-trained language models [28]. Given our rather small training (2,531 observations) and test sets (684 observations), we are satisfied with the performance of the model. We are especially impressed by the model's ability to decode that much information from a given firm website text, such that adding explicit size, age, and sector information does not further improve the model's prediction performance. We also found that our text-only model outperforms an alternative firm characteristics-only model by more than ten percentage points in overall precision, recall, and f1-score. This is especially distinct for the difficult-to-detect product innovator class, where our text-based model achieved a more than twenty percentage points higher recall.

The classification of 685,057 out-of-sample MUP firms resulted in a product innovator probability distribution where most firms have a probability between 10% and 40%. 10.31% of all firms would be classified as product innovators when using the training step classification threshold of 0.5. As there is no reference value for the overall population of firms in Germany, we compared our prediction results to the MIP survey sampling population for which extrapolated population reference numbers are available [19]. In total, 89,372 MUP firms fall into this group of manufacturing and business-oriented service firms with five or more employees. Within this group, a classification threshold of 0.5 would result in a share of 21% predicted product innovators, just short of the surveyed 27%. For the further analysis, we calibrated the classification threshold to 0.401 which results in a share of product innovators that matches the survey benchmark. We also used this threshold to classify MUP firms from sectors and size classes that are not covered in the MIP, given the lack of a better reference values. In the MUP dataset, this shifts the share of predicted product innovators from from 10.31% (0.5 threshold) to 15.12% (0.401 threshold). We continued comparing our prediction results against established innovation indicators at the firm-level (using patent statistics), the regional level (using patent statistics and statistics on R&D personnel), as well as size and sector-levels (using MIP survey extrapolation numbers).

The breakdown by sector showed that the overall sectoral pattern (i.e. the proportions between sectors) is similar to the MIP benchmark. However, our model underestimates the share of product innovator firms in most sectors for which a reference value is available. We attribute this to our model's low recall of the "product innovator" class. Wholesale and ICT services, however, are exceptions, as our model overestimates the share of product innovators in these sectors. Concerning the wholesale sector, we assume that our model is not be able to distinguish between products produced or just sold by a firm. This may lead to a high share of predicted product innovators in the wholesale sector because some assumed product

innovator firms are actually just presenting and selling products of other firms. One possible explanation for the overpredicted share of product innovators in the ICT service sector is that the "tech" sector is nowadays widely considered the sector with the most innovative and future-oriented technologies (with buzz words like Digitalization, Industry 4.0, Internet of Things, Artificial Intelligence and the like). Firms with an innovative agenda or self-concept may use these technologies (or at least the associated buzz words) and mention them on their websites. This could result in a bias in our classification model such that the artificial neural network learns to over-relate these words to innovativeness. ICT firms, which naturally use such "tech" vocabulary on their websites, may then be classified as product innovators too often. A preliminary analysis concerning the importance of word features using SHAP [29] points in exactly this direction and suggest that tech sector affiliated words (e.g. "software", "data", "cloud") may indeed play an important role during the classification. The share of product innovator firms in sectors for which no survey benchmark is available can be considered to be reasonable and indicate that our model may be used for predictions in sectors for which no training data is available. This assumption is supported by our findings on the regional and firm-level (see below). However, we assume that the retail sector, may suffer from overestimated product innovator shares for the same reasons as the wholesale sector.

In conclusion, we suggest to use the raw (continuous) product innovator probabilities for future studies. If a binary indicator is needed, sector-level classification thresholds should be used to cope with the bias we found to be present in our predictions. Given that survey data is available for some sectors, researchers may want to select different classification threshold for each sector such that the predicted share of product innovators matches the shares from the extrapolated survey. Another approach would be to train separate models for each sector.

The breakdown by firm size revealed an interesting, non-linear relationship between firm size and the predicted product innovator probabilities. Over all sectors, the product innovator probability peaks at 500 to 1,000 employees before decreasing until about 2,000 employees. This effect is even more distinct for the exemplary sectors of ICT services and mechanical engineering. The well-known German *Mittelstand* (the bulk of mid-sized and highly innovative German firms) may be visible here. Compared to the MIP survey benchmark, our model almost perfectly predicts the share of product innovators for all size classes, except for very large firms with 1,000 or more employees, where the predictions are clearly below the MIP benchmark. This issue has to be examined in follow-up studies.

At the level of individual firms, we used patent statistics to evaluate our out-of-sample predictions. Considering the raw correlations, we observed a significant Spearman's correlation between our predicted product innovator probabilities and a firm's status as a patent holder (0.29) as well as the number of patents a firm holds (0.25). These correlations remained even after controlling for potential size and sector effects in a regression analysis. They also compare very well to the correlations between a firm's product innovator status from the MIP survey and its status as a patent holder (0.29) and the number of patents the firm holds (0.30). Consequently, we are confident that our predicted innovation indicator is robust even in an out-of-sample setting. However, it has to be noted that patents and the concept of product innovations capture rather different aspects of the innovation system. Patents do not only tend to detect inventions rather than innovation, they are also used as legal protection for technological progress, for example.

The regional patterns of our predictions, with distinct East-West, North-South, and urban-rural trends, are in line with what was reported in the respective literature [30]. We also found very high and significant positive correlations between our predictions and two established innovation indicators available at the regional level (high-tech patent applications and R&D personnel). The microgeographic innovation density maps for Berlin highlighted two further

aspects. First, the results of our prediction model compare very well to the benchmark data from the MIP special survey of Berlin. We identified similar product innovator hotspots in both patterns (city center East and the technology park of Adlershof in the South-East). Again, we assume that the model's bias towards the ICT sector may be the cause for a more pronounced innovation hotspot in the eastern city center, an area with exceptionally high shares of firms from this sector [31]. Second, our predicted innovation indicator can be used to conduct large-scale analysis of regions in any desired geographical resolution, from individual firm locations to aggregated geographical units. The latter can be considered an important contribution because it allows scientist to analyze innovation policies with unprecedented regional and sectoral granularity.

## Conclusion

In this paper, we presented a novel approach on how to predict a highly granular firm-level innovation indicator using deep learning of website texts. We motivated our approach with the need to provide innovation policy making with an innovation indicator that overcomes some of the limitations of traditional indicators from questionnaire-based surveys and patents. Using the website texts of firms included in a traditional innovation survey as training data, we developed an artificial neural network classification model to predict the product innovator probabilities of firms using only their website texts as input. Our choice of training data, as well as our web text selection and preprocessing procedure were intended to allow the neural network to learn from firms' business activity profiles. Eventually, we did not intend to identify distinct product innovations, but firms with business activity profiles that make product innovations very likely. This likelihood (i.e. probability) can be interpreted as a continuous firm-level innovation indicator. The following two research questions were intended to answer the question on the credibility of this new web-based innovation indicator.

### Prediction performance

We concluded that our product innovator classification model achieves a good performance within the test set of firms with a surveyed innovation status. However, we found that the model tends to underpredict the share of product innovators firms, which is reflected in a rather low recall concerning the "product innovator" class. An alternative model that uses only firm characteristics (size, age, and sector) as input, was outperformed by our text-only model by more than ten percentage points f1-score and twenty percentage points recall concerning the difficult-to-detect "product innovator" class. We also found that a combination of web texts and firm characteristics did not improve over a text-only model.

### Patterns from out-of-sample prediction

We predicted product innovator probabilities for 685,057 out-of-sample firms and examined the resulting sectoral, size, regional, and firm-level patterns using MIP survey extrapolations, regional innovation indicators, and firm-level patent statistics.

Compared to the survey extrapolations, we found that our model underestimates the overall share of product innovator firms, if a classification threshold of 0.5 is used (we attribute this to our model's low recall of innovative firms). We subsequently adjusted the classification threshold to a value that produces the same number of innovative firms anticipated by the survey extrapolation benchmark.

The resulting sectoral patterns showed two distinct features. First, the overall trend and proportions between sectors followed the trend anticipated from the survey. Second, we identified a positive bias towards ICT firms, resulting in an overestimated share of product innovators in

this sector which we discussed thoroughly in the Discussion section. Aggregated to size classes, our predictions almost perfectly matched the surveyed benchmarks. Looking at the relation between firm size and the raw product innovator probabilities, we identified a non-linear relationship that may reflect innovative and mid-sized German *Mittelstand* firms.

We also found high and positive correlations between our predictions and firm-level patent statistics (patent counts and patent holder status) that are also robust in a regression setting which controls for sector and size effects. These correlations also matched the correlations we could observe when comparing patent statistics and the original MIP innovation survey results.

The geographical patterns yielded by our model showed very high correlations to regional high-tech patent applications and regional R&D personnel. Lastly, we showed that the micro-geographic prediction patterns match external survey data very well. Overall, we are confident that we created a valuable tool for scientist to analyze innovation at any geographical scale and sectoral level.

## Future research

Future research should concentrate on both the methodological development and the application of our approach. Methodologically, it would be interesting to further investigate which words and word combinations have a significant impact on the neural network's prediction outcome. Additional development of the network's architecture and additional training data, as well as a different preprocessing of the training data, could lead to a better prediction performance. Our proposed approach could also be applied to other target variables from surveys in economics (e.g. process innovators and patent holders) or other fields of social science. Empirical follow-up studies could apply our approach to a wide array of research questions, from innovation policy evaluations to the analysis of knowledge spillovers and technology diffusion. Frequent crawling of firm websites may also allow us to build up a panel database of web-based innovation indicators suitable for time-series analysis.

## Appendix

### Bloat webpage filtering

**Training dataset of bloat and non-bloat webpages.** During web scraping, all texts found on a firm's website are downloaded, regardless of their content and relevance to the study. Alongside valuable web texts describing the firm itself, products, employees, employed technologies, and the like, web texts from imprints, legal information, and HTTP cookie pop-ups are downloaded as well. We also face the problem that our textual data is highly ambiguous in the sense that many websites share common features, e.g. login pages or contact and legal sections. To filter webpages which contain text of mostly unwanted nature (*bloat webpages*), we created a dataset of webpages labeled as either no-bloat (containing unwanted information) or gold (containing relevant information) which we used to train a bloat/no-bloat classification model.

For this purpose, we sampled 10,000 firms from our MUP base dataset and used ARGUS to scrape their websites with a limit of 100 webpages per website and German as the preferred language. We then kept only non-empty webpages written in German (as classified by Python's langdetect library; [32]). From this sample, we drew 10,000 webpages of which 8,080 could be unambiguously labeled as either gold or bloat by hand.

**Bloat webpage filtering results.** Training our classification model with the bloat webpage training data and testing it with a retained part of the bloat webpage data (test set), resulted in a precision, recall, f1-score and support indicated in Table 4. The *precision* score of 0.81

**Table 4. Bloat classification report for test set.**

| label | precision | recall | f1-score | support |
|---|---|---|---|---|
| bloat | 0.81 | 0.48 | 0.61 | 368 |
| no bloat | 0.89 | 0.97 | 0.93 | 1652 |
| avg / total | 0.88 | 0.89 | 0.87 | 2020 |

indicates that the trained model is correct in 81% of cases if the predicted label is "bloat" (i.e. in 19% of cases the prediction is bloat even though the webpage is *no bloat*). Out of all bloat webpages, we identify 48% of webpages correctly (*recall* of 0.48) as being bloat, but fail to detect 52% of bloat webpages. Combining precision and recall by applying a harmonic mean, results in a *f1-score* of 87%. *Support* indicates the respective number of cases. Thus, while having high precision, the recall of the bloat class leaves room for improvements. However, in our case we think it is reasonable to prefer a high precision over high recall, as we primarily want to dismiss webpages that are certainly not relevant.

Based on these findings, we decided to set the threshold of the classification model to 0.9, i.e. we only kept a webpage if the model was highly certain that the webpage is no bloat (probability(no bloat) > 0.9). This filtering step resulted in the exclusion of 309 MIP firms because their websites consisted of bloat webpages only.

## Alternative training data

To assess the adequacy of our training data selection approach, we reran the entire procedure of web scraping, text preprocessing, model training and testing using three alternative datasets. First, we used only the product innovator variable from the 2017 survey (the same survey we used to create our main training dataset) instead of creating our "stable innovator" training dataset. Using this significantly larger dataset (11,506 firm websites) resulted in a f1-score of just 68% in the corresponding test set (see Table 5).

Second, we used the product innovator variable of the more recent MIP of 2018. The results in Table 6 show that this convergence in time of survey data and web data results in a better f1-score, compared to the results using the same survey variable with a one year longer time lag (see Table 5).

Third, we used an alternative product innovator variable of the MIP 2018 which relates directly to the year of the survey. Instead of asking about product innovations introduced by

**Table 5. Product innovator classification report for alternative data test set A.**

| label | precision | recall | f1-score | support |
|---|---|---|---|---|
| non-innovative | 0.72 | 0.83 | 0.77 | 1,784 |
| innovative | 0.62 | 0.47 | 0.53 | 1,093 |
| avg / total | 0.68 | 0.69 | 0.68 | 2,877 |

**Table 6. Product innovator classification report for alternative data test set B.**

| label | precision | recall | f1-score | support |
|---|---|---|---|---|
| non-innovative | 0.75 | 0.88 | 0.81 | 1,264 |
| innovative | 0.61 | 0.38 | 0.47 | 601 |
| avg / total | 0.70 | 0.72 | 0.70 | 1,865 |

**Table 7. Product innovator classification report for alternative data test set C.**

| label | precision | recall | f1-score | support |
|---|---|---|---|---|
| non-innovative | 0.79 | 0.95 | 0.86 | 751 |
| innovative | 0.63 | 0.26 | 0.37 | 258 |
| avg / total | 0.75 | 0.77 | 0.74 | 1,009 |

the firm in the three consecutive years prior to the survey, product innovations in 2018 are surveyed. This survey data, which covers about the same time as the web data scraped in 2018, increases the predictive performance of the model in the corresponding test set even more (f1-score of 0.74; Table 7).

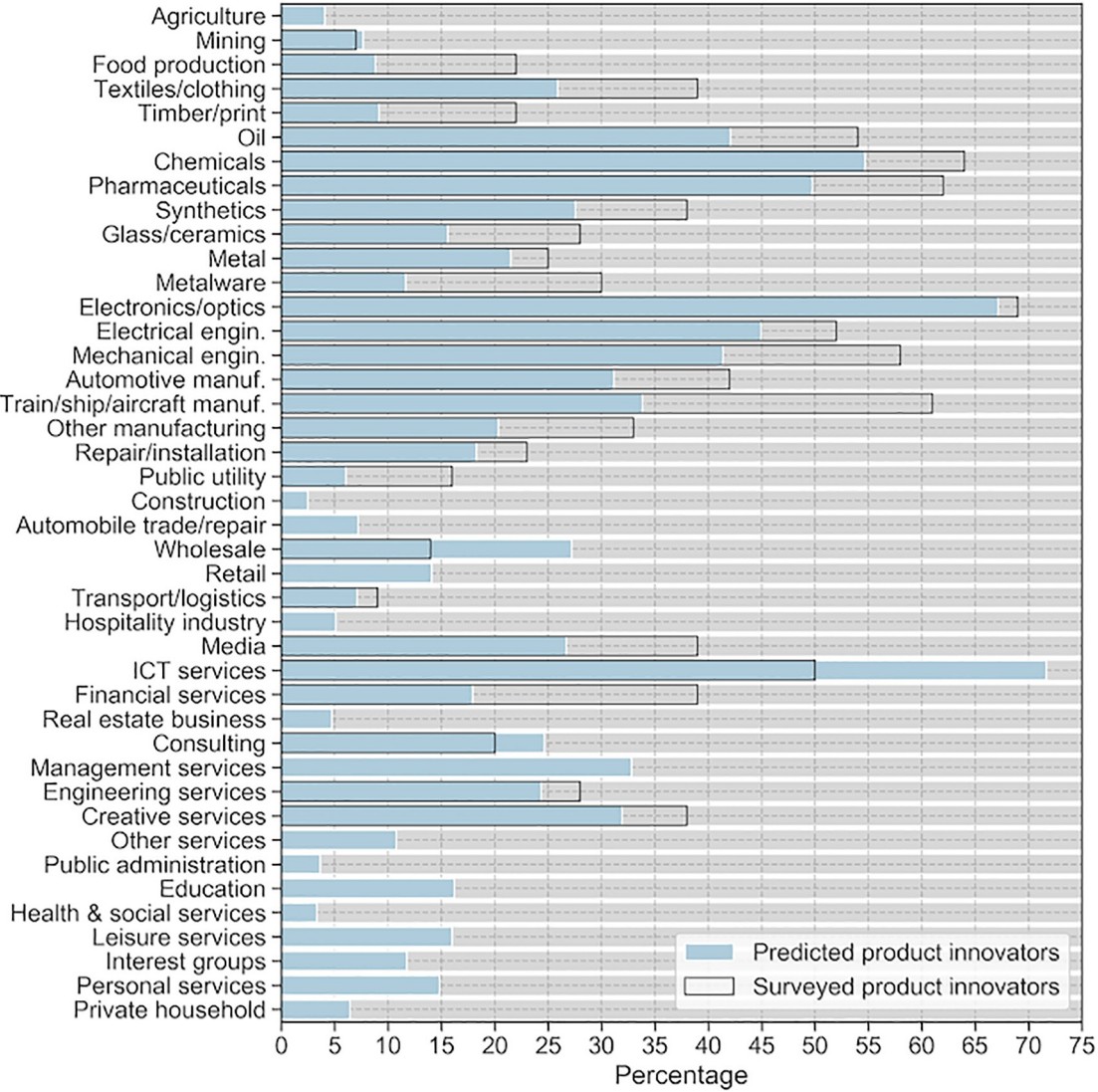

**Fig 4. Predicted product innovator firms by sector.** Share of product innovator firms with five or more employees by sector. Blue bars indicate the predicted shares. Transparent bars indicate extrapolated shares from the MIP innovation survey.

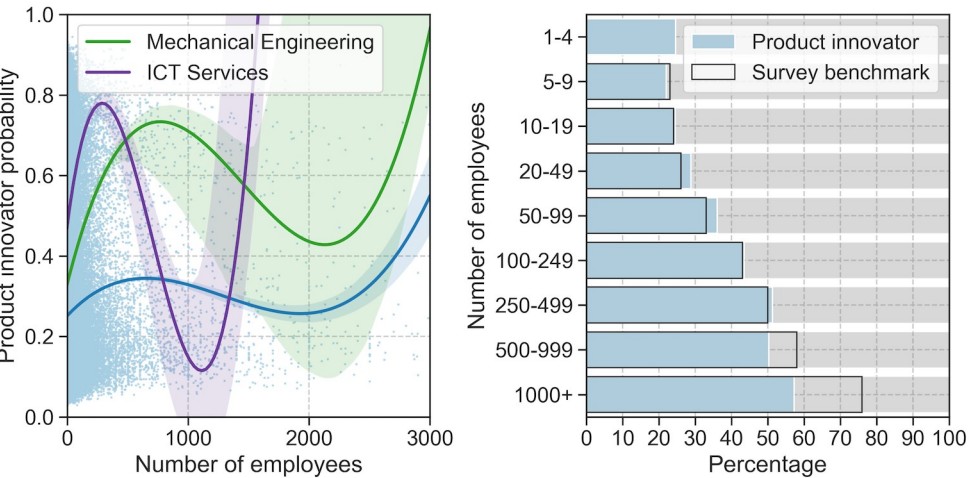

**Fig 5. Predicted product innovator firms by size.** Left panel: Product innovator probabilities by firms size (number of employees) with fitted third order polynomial regression line with 95% confidence interval; all sectors (blue); ICT services (purple), mechanical engineering (green). Right panel: Share of product innovator firms by size classes (predictions in blue; survey extrapolations transparent).

## Share of predicted product innovator firms by sectors and size classes

Fig 4 presents the share of product innovator firms by sector after applying the calibrated classification threshold of 0.401. We also indicate the share of product innovator firms by sector as they are calculated from the MIP questionnaire-based survey (transparent bars) if available for the respective sector. Even though the overall trend and the proportions between sectors are similar to the survey benchmark, underprediction can be seen in all sectors except for wholesale, consulting, and especially ICT firms. It is difficult to assess the predictions scores without a survey benchmark, however, it appears that the ratios between sectors without a benchmark appear to be reasonable. We see very low shares of product innovators in construction and agriculture but higher shares in management services, for example. Accordingly, the ratios between these industries without a comparison score can be considered credible, but we have no way of assessing the accuracy of the individual scores.

Fig 5 shows a breakdown of our predictions by firm size (number of employees). In the left panel, the number of employees is plotted against the predicted product innovator probabilities for all sectors (blue). ICT service (purple) and mechanical engineering firms (green) are plotted as exemplary sectors. It can be seen that the fitted polynomial regression lines of third order indicate a positive non-linear relationship between the size of firms and their product innovator probabilities. The right panel of Fig 5 shows the share of product innovator firms by size groups for sectors covered in the MIP survey (blue) and the corresponding survey extrapolation benchmarks (transparent bars). It can be seen that our predictions match the survey benchmark very well, except for very large firms with more than 1,000 employees where we underestimate the share of product innovators by about 20 percentage points.

## Author Contributions

**Conceptualization:** Jan Kinne, David Lenz.

**Data curation:** Jan Kinne, David Lenz.

**Formal analysis:** Jan Kinne, David Lenz.

**Funding acquisition:** Jan Kinne.

**Investigation:** Jan Kinne, David Lenz.

**Methodology:** Jan Kinne, David Lenz.

**Software:** Jan Kinne, David Lenz.

**Validation:** Jan Kinne, David Lenz.

**Visualization:** Jan Kinne, David Lenz.

**Writing – original draft:** Jan Kinne, David Lenz.

**Writing – review & editing:** Jan Kinne, David Lenz.

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
