## [Decision Letter · Decision Letter 0]

10 Dec 2020

PONE-D-20-25015

Predicting innovative firms using web-mining and deep learning

PLOS ONE

Dear Dr. Lenz, 

Thank you for submitting your manuscript to PLOS ONE. After careful consideration, we feel that it has merit but does not fully meet PLOS ONE’s publication criteria as it currently stands. Therefore, we invite you to submit a revised version of the manuscript that addresses the points raised during the review process.

The reviewer and I find your topic is valuable, timely, and interesting. However, there are a few limitations that need clarification and revision. For example, the sectoral performance of model needs revision and clarification since it is counter-intuitive. 

We look forward to receiving your revised manuscript.

Kind regards,

Wonjoon Kim, Ph.D

Academic Editor

PLOS ONE

We note that one or more of the authors are employed by a commercial company: istari.ai.

2.1. Please provide an amended Funding Statement declaring this commercial affiliation, as well as a statement regarding the Role of Funders in your study. If the funding organization did not play a role in the study design, data collection and analysis, decision to publish, or preparation of the manuscript and only provided financial support in the form of authors' salaries and/or research materials, please review your statements relating to the author contributions, and ensure you have specifically and accurately indicated the role(s) that these authors had in your study. You can update author roles in the Author Contributions section of the online submission form.

2.2. Please also provide an updated Competing Interests Statement declaring this commercial affiliation along with any other relevant declarations relating to employment, consultancy, patents, products in development, or marketed products, etc.  

4. We note that Figure 3 in your submission contain map images which may be copyrighted. All PLOS content is published under the Creative Commons Attribution License (CC BY 4.0), which means that the manuscript, images, and Supporting Information files will be freely available online, and any third party is permitted to access, download, copy, distribute, and use these materials in any way, even commercially, with proper attribution. For these reasons, we cannot publish previously copyrighted maps or satellite images created using proprietary data, such as Google software (Google Maps, Street View, and Earth). For more information, see our copyright guidelines: http://journals.plos.org/plosone/s/licenses-and-copyright.

4.1.    You may seek permission from the original copyright holder of Figure 3 to publish the content specifically under the CC BY 4.0 license. 

4.2.    If you are unable to obtain permission from the original copyright holder to publish these figures under the CC BY 4.0 license or if the copyright holder’s requirements are incompatible with the CC BY 4.0 license, please either i) remove the figure or ii) supply a replacement figure that complies with the CC BY 4.0 license. Please check copyright information on all replacement figures and update the figure caption with source information. If applicable, please specify in the figure caption text when a figure is similar but not identical to the original image and is therefore for illustrative purposes only.

Reviewers' comments:

Reviewer's Responses to Questions

**Comments to the Author**

1. Is the manuscript technically sound, and do the data support the conclusions?

Reviewer #1: Yes

2. Has the statistical analysis been performed appropriately and rigorously? 

Reviewer #1: Yes

3. Have the authors made all data underlying the findings in their manuscript fully available?

Reviewer #1: No

4. Is the manuscript presented in an intelligible fashion and written in standard English?

Reviewer #1: Yes

5. Review Comments to the Author

Reviewer #1: This paper utilizes firms' websites with deep neural network methodology to predict firms' product innovation. By doing so, it provides timely and cost-effective way of predicting firms' product innovation.

The method is based on systematic scientific procedures and its overall performance is found to be acceptable.

However, its performance at sectoral level is counter-intuitive.

The authors wrote that the model under-estimate the share of product innovators in sectors for which a reference value is available but the share of product innovators for which no training data is available can be considered to be reasonable.

It would be reasonable to think that a model's performance is better for the sector with a reference value than the sector without a reference value. If the converse is true, the less you have data with a reference value, the better the model's performance would be.

The authors need to explain the sectoral performance of model in a different way.

6. PLOS authors have the option to publish the peer review history of their article (what does this mean?). If published, this will include your full peer review and any attached files.

Reviewer #1: No

---

## [Author Response · Author response to Decision Letter 0]

25 Jan 2021

Dear Reviewer,

 Thank you very much for the positive review. The author team has made every effort to improve all the formal errors and ambiguities in our manuscript that you mentioned. We sincerely hope that our revised manuscript will meet your requirements.

Comment 1: The authors wrote that the model under-estimate the share of product innovators in sectors for which a reference value is available but the share of product innovators for which no training data is available can be considered to be reasonable. It would be reasonable to think that a model's performance is better for the sector with a reference value than the sector without a reference value. If the converse is true, the less you have data with a reference value, the better the model's performance would be. The authors need to explain the sectoral performance of model in a different way. 

Response 1: We fully agree with this keen observation. It is true that we are causing confusion for the reader here through an unclear choice of words. We have adjusted the corresponding passage in the manuscript. The corresponding passage has been marked in red in the manuscript. We hope that the new wording allows a clearer interpretation of our results.

Sincerely yours, 

the team of authors

---

## [Decision Letter · Decision Letter 1]

11 Mar 2021

Predicting Innovative Firms using Web Mining and Deep Learning

PONE-D-20-25015R1

Dear Dr. Lenz,

We’re pleased to inform you that your manuscript has been judged scientifically suitable for publication and will be formally accepted for publication once it meets all outstanding technical requirements.

Kind regards,

Wonjoon Kim, Ph.D

Academic Editor

PLOS ONE

Reviewers' comments:

Reviewer's Responses to Questions

**Comments to the Author**

1. If the authors have adequately addressed your comments raised in a previous round of review and you feel that this manuscript is now acceptable for publication, you may indicate that here to bypass the “Comments to the Author” section, enter your conflict of interest statement in the “Confidential to Editor” section, and submit your "Accept" recommendation.

Reviewer #1: All comments have been addressed

2. Is the manuscript technically sound, and do the data support the conclusions?

Reviewer #1: Yes

3. Has the statistical analysis been performed appropriately and rigorously? 

Reviewer #1: Yes

4. Have the authors made all data underlying the findings in their manuscript fully available?

Reviewer #1: Yes

5. Is the manuscript presented in an intelligible fashion and written in standard English?

Reviewer #1: Yes

6. Review Comments to the Author

Reviewer #1: All comments have been addressed.

The authors have adjusted the corresponding passage in the manuscript.

7. PLOS authors have the option to publish the peer review history of their article (what does this mean?). If published, this will include your full peer review and any attached files.

Reviewer #1: No

---

## [Editor Report · Acceptance letter]

16 Mar 2021

PONE-D-20-25015R1 

Predicting Innovative Firms using Web Mining and DeepLearning 

Dear Dr. Lenz:

I'm pleased to inform you that your manuscript has been deemed suitable for publication in PLOS ONE. Congratulations! Your manuscript is now with our production department. 

Kind regards, 

on behalf of

Dr. Wonjoon Kim 

Academic Editor

PLOS ONE